# ZCTG: A Zero-Shot Framework for Automatic Video Chaptering and Title Generation

## Abstract

In the vast landscape of video content, breaking down lengthy videos into chapters accompanied by concise, descriptive titles greatly enhances searchability and retrieval efficiency. While recent advancements in this field often incorporate multiple data modalities along with human-annotated chapter titles, access to such data, like speech transcripts or audio, is not always guaranteed. Moreover, the manual annotation of chapter titles is expensive and time-consuming. To address these challenges, we introduce ZCTG, a novel and unified zero-shot framework designed to generate video chapters and their concise titles for untrimmed videos. ZCTG utilizes the combined capabilities of scene graphs and Large Language Models (LLMs). The advantages of ZCTG are three-fold: 1) offers practical utility, relying solely on video data; 2) eliminates the need for detailed chapter title supervision; 3) exhibits excellent generalization capabilities in a completely zero-shot setting, without any training needed. We conduct an extensive evaluation on VidChapters-7M and GTEA datasets, which include videos of varying duration and domains, to demonstrate the efficacy of our proposed framework.

## 1 Introduction

In today's digital landscape, where online content serves diverse purposes such as marketing, tutorials, entertainment, etc., across multiple platforms, there has been a remarkable surge in the consumption of video content. Yet, sifting through this vast array of videos can pose a challenge to users, often overwhelming them, leading to a suboptimal user experience. Segmenting videos into smaller chunks with concise, descriptive titles can significantly improve content accessibility, navigation, and overall user experience. This process, known as video chapter generation, involves dividing a video into segments based on its content and creating titles that accurately reflect each part.

This task is closely related to video captioning, where the goal is to provide detailed descriptions (captions) for a given video, capturing all events/scenes detectable by an algorithm. On the other hand, in video chapter generation, the focus is on partitioning a video into segments (chapters), each with some notion of internal temporal coherence, and then crafting concise titles that summarize semantic highlights of the chapters. Hence, while existing dense video captioning techniques Krishna et al. (2017); Wang et al. (2021); Zhou et al. (2018); Zhu et al. (2022) may yield impressive results for generating detailed descriptions for a video, they are not directly suitable for our task.

Video platforms like YouTube provide users with the option to manually add timestamps and titles for video chapters. However, this manual process can become increasingly challenging, especially for longer videos. To address this, efforts have been made to automate this task, as demonstrated in works such as Cao et al. (2022); Yang et al. (2024), but the field remains relatively underexplored. Such methods utilize both video content and Automatic Speech Transcripts (ASR) or audio and require chapter title annotations for training. However, the availability of ASR data may be limited across various video categories, posing a challenge to the performance of multimodal frameworks. While we acknowledge the importance of multi-modal supervision in such challenging tasks, we argue that a framework that takes only videos as input and generates chapter titles in a zero-shot setting can mitigate these limitations.

In this paper, we introduce **Z**ero-Shot Video **C**hapter **T**itle **G**enerator (**ZCTG**), a unified, novel framework designed to generate chapter titles for video content without relying on annotated data (chapter titles) or additional input modalities typically required during training in existing methods.

The zero-shot nature of our framework also eliminates the need for any task-specific training/fine-tuning, thereby enhancing its generalizability across diverse video types and domains. Unlike conventional methods that require pre-existing annotations or multimodal data such as text or audio inputs, ZCTG operates solely on video frames, leveraging visual information to comprehend the underlying content and generate chapters. We employ scene graph representation and Large Language Models (LLMs) to generate concise titles for each video chapter, capturing its essence effectively. To the best of our knowledge, ZCTG is the first unified framework designed for automatic video chapter and title generation in a zero-shot scenario. We evaluate the performance of ZCTG using two diverse datasets: the GTEA dataset Fathi et al. (2011), which focuses on daily cooking videos captured in controlled environments, and the VidChapters-7M dataset Yang et al. (2024), which consists of a large collection of videos of varying lengths and subjects sourced from YouTube. Our experimental results demonstrate the effectiveness of ZCTG in generating informative, relevant chapter titles in a zero-shot setting.

## 2 RELATED WORK

The video chapter generation task comprises two primary stages: first, the temporal segmentation of the video into distinct chapters, and the generation of a natural language title for each chapter. Therefore, video chapter generation intersects with various other video-based tasks such as video shot detection Rui et al. (1998); Sidiropoulos et al. (2011), temporal action localization Chao et al. (2018); Cheng & Bertasius (2022); Shou et al. (2016), temporal action segmentation Farha & Gall (2019); Sarfraz et al. (2021); Li et al. (2021b) and many more. However, the task of video chapter title generation differs from these other tasks because it involves creating natural language titles for each video chapter.

Temporal action segmentation methods require capturing the long-range dependencies across the video to create segments of actions. Prior research has introduced temporal and dilated convolutional networks as solutions to capture these dependencies Lea et al. (2017); Lei & Todorovic (2018); Farha & Gall (2019); Huang et al. (2020); Ishikawa et al. (2021); Li et al. (2020); Wang et al. (2020). However, these approaches typically depend on annotated datasets, which are resource-intensive to acquire. Consequently, the field has witnessed a growing interest in weakly supervised and unsupervised methods as a mitigation to these challenges Sarfraz et al. (2021); Chang et al. (2019); Ding & Xu (2018); Huang et al. (2016); Kuehne et al. (2018).

While temporal action segmentation can identify similar events throughout lengthy videos, navigating such content without the aid of natural language titles can be challenging, particularly for long videos. The annotation of video chapters with concise titles can facilitate automated navigation of the content. In this context, the video chapter title generation task has relevance to other caption generation tasks such as video captioning Gao et al. (2017); Lin et al. (2022); Luo et al. (2020); Pan et al. (2017); Wang et al. (2018); Zhang et al. (2020b), video title generation Zhang et al. (2020a); Zeng et al. (2016); Amirian et al. (2021), and dense video captioning tasks Krishna et al. (2017); Wang et al. (2021); Zhou et al. (2018); Zhu et al. (2022). Some of the recent and notable efforts in video caption and description generation tasks are VideoLLaMA Zhang et al. (2023) and Intern-Video2 Wang et al. (2024). However, these frameworks exhibit certain limitations, such as their inability to capture temporal relationships in long videos, leading to the generation of erroneous titles. Furthermore, they lack the capability to detect chapters within lengthy videos. Thus, there is a pressing need for frameworks capable of automating chapter and its title generation for any video, thereby minimizing manual effort.

The concept of video chapter title generation has been defined and studied by Yang et al. (2024) in their work. It was observed that models trained on visual and ASR (Automatic Speech Recognition) data outperformed those trained solely on visual data. Cao et al. (2022) employ a multi-modal feature extraction method using video content and narration text to localize the video segments (chapters) and generate titles in a supervised manner. However, the availability of ASR or other data modalities as well as fine-grained annotations may be limited.

Hence, we present a zero-shot framework for video chapter title generation that eliminates the requirement for multiple data modalities and training using densely annotated large datasets. Our proposed approach utilizes only video content for chapter and its title generation and combines the benefits of scene graph representation alongside the generative capabilities of Large Language Mod-

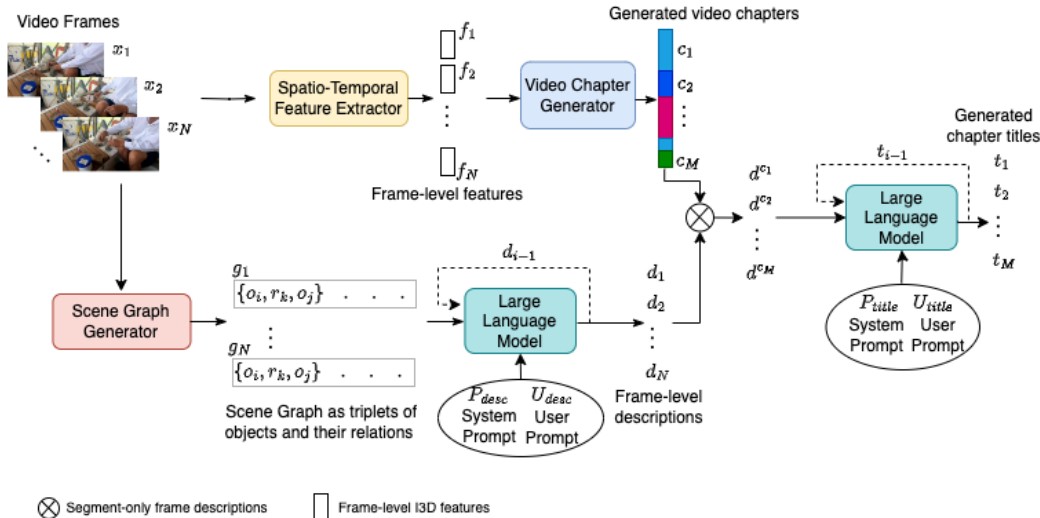

Figure 1: ZCTG - Overview of the proposed framework.

els (LLMs). Owing to its zero-shot nature, this framework has wide applicability across videos of varying lengths and genres, thereby enhancing its versatility.

## 3 METHODOLOGY

### 3.1 PROBLEM STATEMENT

Given a sequence of video frames $X = \{x_1, x_2, ..., x_N\}$, where $N$ represents the total number of frames, our objective is to identify contiguous segments that encapsulate distinct actions in terms of semantics and their titles describing the content in it. These segments are referred to as video chapters, denoted by $C = \{c_1, c_2, ..., c_M\}$, with $M$ being the total number of video chapters. The chapters are associated with chapter titles denoted by $T = \{t_1, t_2, ..., t_M\}$. Since this is a zero-shot setting, no information about the ground truth (video chapter boundaries or chapter titles) is available, and no training has been performed using $X$.

### 3.2 ZCTG: PROPOSED FRAMEWORK

We propose the ZCTG framework for automatic video chapter and title generation, comprising of two primary tasks: *Video Chapter Generation* and *Chapter Title Generation*. Figure 1 depicts the overall framework of ZCTG. The top pipeline of the framework generates video chapters using the spatio-temporal video frame features. The lower pipeline generates the titles for the chapters that capture the content of the respective video chapters. To achieve this, the visual content is converted to text representation using scene graphs which is then given to a Large Language Model (LLM).

#### 3.2.1 VIDEO CHAPTER GENERATION

For generating semantically relevant chapter titles, creating meaningful video chapters is essential. In order to generate meaningful video chapters, it is essential to consider both spatial and temporal content. Hence, we use Spatio-Temporal Feature Extractor, which extracts spatio-temporal features at the frame level using a pre-trained I3D Wang et al. (2019), a robust 3D convolutional neural network represented by $F(.)$. To extract features for a video frame $x_i$, we incorporate its neighboring frames within a window size of $2p + 1$, as illustrated in Equation 1.

$$f_i = F\left( \overset{i+p}{\underset{j=i-p}{\|}} x_j \right) \tag{1}$$

where $p$ is the number of frames to be considered before/after $x_i$. This sliding window method ensures that the extracted features encompass spatial and temporal information, essential for producing precise video segments.

Once the spatio-temporal features are extracted for all frames, they are fed into a Video Chapter Generator module. It consists of a model designed to segment the video into chapters based on their content. For this step, we employ an off-the-shelf, unsupervised temporal action segmentation technique, TW-FINCH Sarfraz et al. (2021). We choose this unsupervised algorithm as we do not assume the availability of fine-grained labels. The generated chapters are based on internal temporal coherence derived from spatio-temporal frame features, which may typically differ from standard video shot changes. While shot changes focus more on scene changes, the Video Chapter Generator captures subtle variations within the scene more effectively.

Typically, TW-FINCH requires predefining the number of clusters. However, we refrain from assuming any prior knowledge about the number of segments or activities in a video. Considering that natural videos usually contain around 10-15 actions on average, we set the number of clusters $K = 10$ for all our experiments unless specified otherwise. Let $D(.)$ represent our Video Chapter Generator, then,

$$C = D\left(\left(\overset{N}{\underset{j=1}{\|}} f_j\right), K\right), \tag{2}$$

$C = \{c_1, c_2, ..., c_M\}$, where $C$ is the generated video chapters and $M$ denotes the total number of chapters. Note that $M \geq K$, as the same action may occur at multiple time points within a video.

### 3.2.2 CHAPTER TITLE GENERATION

Once the video chapters $C$ are generated, the next task is to create descriptive titles for each chapter. Unlike existing methods Yang et al. (2024); Cao et al. (2022), which rely on both audio speech transcripts (ASRs) with video data, our framework offers a novel alternative using visual data only. This is particularly beneficial, as it removes the dependency on ASR for every video. The key challenge lies in translating visual content into meaningful textual representations that effectively capture both spatial and temporal cues. To address this, we make use of the expressive potential of scene graphs. We choose scene graph representation as it captures the interactions among various objects, thereby facilitating the understanding of the scene dynamics.

For every frame, we first extract its scene graph representation using the Scene Graph Generator module, $A(.)$. We use a pre-trained scene graph generation module Li et al. (2021a) as our $A(.)$. The scene graph is expressed as a group of triplets $\{o_i, r_k, o_j\}$, where $o_i$ and $o_j$ denote objects within the frame, and $r_k$ signifies the relation between them. For every frame $x_i$, its corresponding scene graph $g_i$ is extracted as $g_i = A(x_i)$, where $g_i \in \mathbb{R}^{Q \times 3}$ and $Q$ represents the number of triplets. Li et al. (2021a) considers the most confident 80 object predictions and derives all pairwise relations among them. However, considering all $(80 \times 80)$ relations poses several challenges - first, less confident relations may introduce irrelevant noise, which will affect the quality of generated chapter titles; second, it increases the computational complexity in subsequent stages of the pipeline; and lastly, the inclusion of all relations will be limited by the fixed input token size of the LLM. Hence, we employ a two-step filtering mechanism to select the most confident $Q$ triplets, aiming to mitigate these challenges. First, we select the $Q$ most confident predicted objects, followed by considering only the $Q$ most confident relations among these selected objects. We set $Q = 10$ empirically and refer to A.1 for the corresponding experiments. This filtering minimizes the inclusion of noisy predictions and is in the token limit of the LLM, to be used in later stages.

Even though the scene graphs for frames, $G = \{g_1, g_2, ...\}$ convert the visual content in textual form to be given as input to LLM, this presents several challenges - the scene graph triplets contain information consisting solely of objects and their relations without any additional context; directly aggregating all the triplets from the frames of a video chapter will not represent meaningful spatial and temporal cues. Hence, we propose a novel two-step solution to tackle these challenges.

First, we leverage the contextual capabilities of LLMs to generate concise descriptions using $g_i$ for each frame. This will provide the necessary context missing in the scene graph triplets. To create a frame description $d_i$, the LLM, $L(.)$ is provided with a system prompt $P_{desc}$, a user prompt $U_{desc}$ combined with the current frame's scene graph triplets $g_i$, and the generated description for

the previous frame $d_{i-1}$. We incorporate $d_{i-1}$ to introduce temporal context during description generation. For the first frame, we set $d_{i-1}$ as 'First Frame'.

After generating descriptions for each frame, the next step involves using them to generate chapter titles $T = \{t_1, .., t_M\}$. For each chapter $c_i$, the title $t_i$ is generated using the LLM $L(.)$, with input of a system prompt $P_{title}$ and a user prompt $U_{title}$ combined with the preceding segment's generated title $t_{i-1}$ and frame descriptions $G^{c_i}$ for all frames $X^{c_i}$ within segment $c_i$ (refer to Equation 3). Similar to the previous step, the inclusion of $t_{i-1}$ is employed to maintain temporal coherence and consistency. We handle $t_{i-1}$ for the first frame similar to frame descriptions by setting it as 'Start of the video'.

$$t_i = L(P_{title}, U_{title} \| \left( \overset{R}{\underset{j=1}{\|}} d_j \right) \| t_{i-1}), \quad i = 1, 2, .., M \tag{3}$$

where $R$ is the number of frames in a chapter and $M$ is the total number of chapters.

Dividing the chapter title generation process into two steps, frame description generation followed by title generation, offers several advantages - it augments scene graph representation with additional context and ensures temporal consistency in the generated titles. We summarize the steps in our framework ZCTG in Algorithm 1.

---

**Algorithm 1** : ZCTG

---

1: **Input:** Video frames X$=\{x_1, \ldots, x_N\}$, $F$ (I3D), $D$ (Video Chapter Generator), $A$ (Scene Graph Generator), $L$ (LLM), $P_{desc}, U_{desc}$ (System and user prompt for frame description), $P_{title}, U_{title}$ (System and user prompt for chapter title generation)
2: **Output:** Video Chapters $C = \{c_1, \ldots, c_M\}$, Chapter Titles $T = \{t_1, \ldots, t_M\}$
3: **Inference Strategy:**
4: $f = F(X)$                     ▷ Extract spatio-temporal frame features
5: $C = D(f)$                         ▷ Generate video chapters
6: T = {}
7: **for** $c_i$ in $C$ **do**
8:     **for** $x_j$ in $X^{c_i}$ **do**
9:         $g_j = A(x_j)$                   ▷ Generate scene graph
10:         $d_j = L(g_j, P_{desc}, U_{desc}, g_{j-1})$       ▷ Generate frame description
11:     $t_i = L(\|_{k=1}^{|X^{c_i}|} d_k, P_{title}, U_{title}, t_{i-1})$     ▷ Generate video chapter title
12:     T.append($t_i$)
13: **return** $C, T$

---

## 4 EXPERIMENT RESULTS

In this section, we outline the experimental settings for conducting our experiments. Following this, we discuss the evaluation results on VidChapters-7M and GTEA and discuss other analysis experiments as well.

### 4.1 EXPERIMENT SETTINGS

#### 4.1.1 DATASETS

For the evaluation of ZCTG, we use the VidChapters-7M Yang et al. (2024) and GTEA Fathi et al. (2011) datasets. The VidChapters-7M dataset comprises 817,076 videos along with their chapter titles. The chapter titles are annotated by users, as the dataset is curated from YouTube and selectively filtered to include only those videos with user-annotated chapter titles. These videos encompass a diverse array of domains, including education and instructional content. On an average, a video lasts 1354 seconds. The dataset is partitioned into 801,000 training videos, with 8,200 each for validation and testing. We report results on the test set, which consists of 6,762 videos (downsampled to 1 FPS) due to some videos being inaccessible or requiring special permissions for access.

The GTEA dataset comprises 28 egocentric videos featuring 7 distinct cooking activities, such as preparing coffee and making a sandwich, conducted by 4 unique subjects. This dataset has 11 sub-

actions annotations, including background. We utilize all 28 videos (1 frame sampled out of every 10 frames) from this dataset for our evaluation.

### 4.1.2 NETWORKS

Spatio-temporal feature extractor: We utilize a pre-trained I3D network as the spatiotemporal feature extractor $F(.)$. The code and pre-trained model can be accessed here. For each frame, we extract a 1024-dimensional spatio-temporal feature solely from the RGB input.

Video chapter generator: To segment videos into chapters, we utilize TW-FINCH Sarfraz et al. (2021) for its strong performance in unsupervised temporal segmentation. The implementation provided by the authors[1] is used in our experiments, with $K = 10$ as the default setting unless specified otherwise. Additionally, we explore alternative temporal segmentation techniques and experiment with value of $K$, which is discussed in Section 4.2.

Large langauge models (LLMs): To generate frame-level descriptions and chapter titles, we utilize the Vicuna v1.5 (13B) model Zheng et al. (2023), which contains 13 billion parameters and supports a context length of up to 16,000 tokens. Built on the Llama 2 architecture, Vicuna v1.5 is fine-tuned using user conversations from ShareGPT. We also explore other LLM models in our experiments, discussed further in Section 4.2. Refer to A.6 for details about the prompts used for ZCTG.

Baseline using Video-LLaMA: Since existing baselines do not directly align with our proposed framework, we use Video-LLaMA Zhang et al. (2023) (based on finetuned Llama 2 (7B) model[2]) as a reference point. While Video-LLaMA demonstrates excellent performance in generating video and image descriptions, it lacks a dedicated chapter generation module. To ensure a fair comparison, we adapt the Video-LLaMA framework to incorporate chapter creation and title generation functionalities. In line with our proposed method, we employ a pretrained I3D network as the spatiotemporal feature extractor to extract frame embeddings and generate video chapters ($K = 10$). Each segmented chapter is then fed into the Video-LLaMA for title generation. Further details on the textual prompts used for this task can be found in A.7.

### 4.1.3 EVALUATION METRICS

Considering the multimodal nature of our problem, which involves both videos and generated textual titles, we evaluate our proposed approach, ZCTG using a range of metrics.

Vision-Language metrics: We use CLIPScore (CS) Hessel et al. (2021) to measure the similarity between the frames of the video chapter and its generated title. The CLIPScore ranges between 0 to 100 and calculated using Torchmetrics library Nicki Skafte Detlefsen et al. (2022).

Language metrics: We also report purely language-based metrics i.e. comparing generated titles with the ground-truth titles. We report BLEU (B$n$) Papineni et al. (2002) where $n = \{1, 2, 3, 4\}$ is the n-gram value, and METEOR (M) Banerjee & Lavie (2005). Following Yang et al. (2024), we also report SODA_c (S) Fujita et al. (2020) for overall evaluation as it first finds optimal matching of the generated chapters with the ground-truth ones and then calculates METEOR scores for the titles. The F-scores are then calculated to penalize the redundant chapters.

Video chapter generation metrics: To evaluate the chapters generated by Video Chapter Generator module, we use two metrics - Mean over Frames (MoF) and Intersection over Union (IoU). Following the evaluation of TW-FINCH Sarfraz et al. (2021), we perform Hungarian matching of the generated chapters and ground-truth chapters for calculating these metrics.

LLM-based metrics: Given the exceptional ability of LLMs to understand the context of the generated text, we also evaluate our method using a Judge LLM. Inspired by evaluation criteria used in Maaz et al. (2023), we evaluate three aspects of the generated titles (on a scale of 0-5):

    i. Contextual understanding: Assessing if the generated titles capture the overall context of the video and its chapters.

    ii. Temporal understanding: Gauging how well the generated titles grasp the temporal sequence of events happening in the video.

---

[1] https://github.com/ssarfraz/FINCH-Clustering/tree/master/TW-FINCH
[2] https://huggingface.co/DAMO-NLP-SG/Video-LLaMA-2-7B-Finetuned

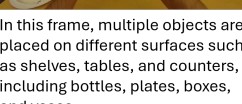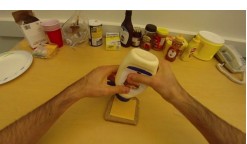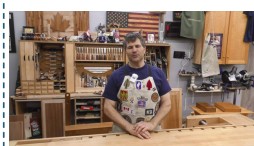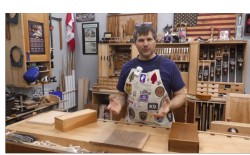

In this frame, multiple objects are placed on different surfaces such as shelves, tables, and counters, including bottles, plates, boxes, and vases.

In this frame, a person is pouring liquid from a bottle into a cup on the table, while another bottle is on the counter and a bag is on the floor.

In this frame, a man wears a shirt and glasses, holds a box on a shelf, and has hair, a head, arms, and hands.

A man wearing glasses and a shirt interacts with various objects, including a box on a counter and a book.

Figure 2: Frame descriptions generated by ZCTG for videos from GTEA (left) and VidChapters-7M (right).

    iii. Correctness of information: Verifying how accurate the generated titles are.

For this evaluation, we utilize the ChatGPT-3.5 model. We minimally adapt the prompts from Maaz et al. (2023) to suit our specific task of video chapter title generation. Details about the prompts and the evaluation process can be found in A.2.

For the VidChapters-7M dataset, ground-truth chapter titles are provided. Since our proposed method follows a fully zero-shot scenario not having any form of supervision, the generated chapters and their titles may differ from the ground-truth. In these instances, we compute the evaluation metrics as follows: for each ground-truth segment, we treat all generated titles by ZCTG as predictions to be compared against the ground-truth title and calculate the evaluation metric. Refer to A.2 for examples. In the case of the GTEA dataset, where ground-truth titles are not available, we report the CLIPScore (CS) only.

## 4.2 Results and Discussion

Chapter title generation: To yield chapter titles, we begin by generating frame descriptions using the visual information represented using scene graphs. Figure 2 showcases the descriptions produced by ZCTG for frames at different timestamps from the GTEA and VidChapters-7M datasets. These descriptions depict the scene and effectively capture the ongoing activities. For example, in the second column, the description accurately recognizes the person squeezing the sauce, identifying the objects in view and their interactions, such as 'pouring liquid'. These descriptions play a key role in generating precise chapter titles.

Table 1: Evaluation results using Vision-Language and Language metrics on VidChapters-7M Dataset. *Numbers are quoted from Yang et al. (2024).

| Method | Modalities | CS | B1 | B2 | B3 | B4 | M | S |
|---|---|---|---|---|---|---|---|---|
| Vid2Seq* | Visual+Speech | - | 0.1 | 0.0 | 0.0 | 0.0 | 0.1 | 0.1 |
| Ours | Visual | 20.90 | 0.24 | 0.00 | 0.00 | 0.00 | 0.03 | 4.1 |

We present the evaluation results of ZCTG on the VidChapters-7M dataset in Table 1. The results for Vid2Seq Yang et al. (2023), originally proposed for dense video captioning, are quoted from Yang et al. (2024) and it is pretrained on C4 and Howto100M datasets and uses visual and speech data modalities. As videos in VidChapters-7M dataset are typically long, we perform an additional step of summarizing the frame descriptions after an uniform interval (20 frames) to address resource constraints. The details about this step can be found in A.3. Notably, ZCTG outperforms or achieves results comparable to the baseline, Vid2Seq Yang et al. (2023), despite relying solely on the videos, unlike Vid2Seq, which uses multiple modalities.

Additionally, we assess our baseline, VideoLlama, on this dataset. However, a limitation of VideoLlama is its ability to handle very long videos. Due to this limitation and resource constraints, we report metrics only on a subset of VidChapters-7M. In this subset, ZCTG surpasses VideoLlama with a CLIPScore of 21.3, compared to VideoLlama's score of 17.30. Details on the experimental setup and these results are available in A.4.

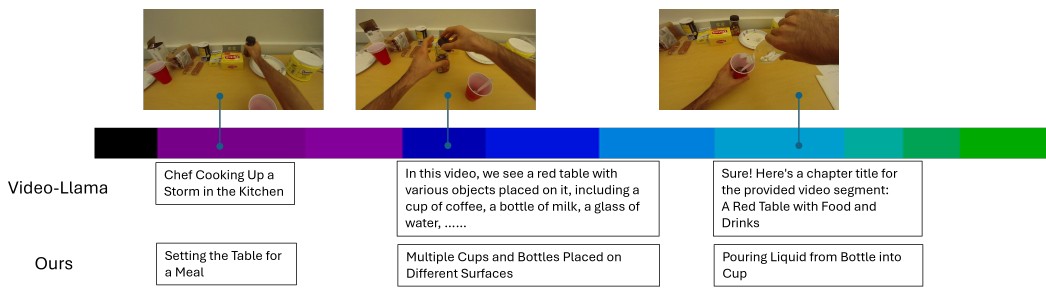

GT: Intro | What is minimal aesthetic | Color | Shapes | Accessories

Input Frames

Ours: Man Dressing Up and Interacting with Objects | Man Getting Dressed for the Day | Man Changing Outfit and Accessories | Man Accessorizing and Interacting with Objects

Figure 3: Generated video chapters and their titles by ZCTG and ground-truth for a video about minimal aesthetic from VidChapters-7M dataset.

Video-Llama: Chef Cooking Up a Storm in the Kitchen | In this video, we see a red table with various objects placed on it, including a cup of coffee, a bottle of milk, a glass of water, ...... | Sure! Here's a chapter title for the provided video segment: A Red Table with Food and Drinks

Ours: Setting the Table for a Meal | Multiple Cups and Bottles Placed on Different Surfaces | Pouring Liquid from Bottle into Cup

Figure 4: Generated video chapters and their titles by VideoLlama (baseline) and ZCTG for a video of making coffee from GTEA dataset.

Figure 3 is an example of chapters and their corresponding titles generated for a video about minimal aesthetics from the VidChapters-7M dataset. The generated titles closely align with the visual content (capturing events like changed outfits and accessorizing), while the ground-truth titles show less coherence (such as 'shape' and 'color') with both the generated titles and the visual content. This discrepancy explains the low language metric scores, which are generally based on n-gram comparisons. However, the low scores do not imply that the generated titles are inaccurate. As a matter of fact, they effectively capture the underlying semantics of the video chapters. Refer A.8 for more such examples.

Table 2: Evaluation results on GTEA Dataset.

| Method | CS |
|---|---|
| Video-Llama | 19.42 |
| Ours | 25.40 |

The evaluation results for the GTEA dataset are summarized in Table 2. We report only the CLIPScore for this dataset, as other metrics depend on ground-truth titles, which are unavailable. Our results indicate that ZCTG significantly outperforms the VideoLlama baseline. The titles generated by VideoLlama, illustrated in Figure 4, are often neither concise nor well-aligned with the visual content. For instance, it inaccurately describes a yellow table as a 'red table'. On the other hand, the chapter titles produced by ZCTG are highly aligned with the visual content, effectively capturing events such as 'pouring liquid from a bottle'.

LLM-based evaluation: Table 3 shows the results obtained using ChatGPT 3.5 as the Judge LLM on the VidChapters-7M dataset. These results reveal that the chapter titles generated by ZCTG are contextually rich, supporting our earlier experiments and observations. Similar to language metrics, the generated titles are evaluated against the ground truth. As discussed previously, ground truth titles do not always show a strong correlation with the visual content. It may be one of the reasons why the scores for correctness of information and temporal understanding are lower. We will address potential improvements in these areas in future work.

Influence of different LLM models: The LLM is a fundamental component of ZCTG. These models are pretrained on large-scale datasets. To examine their effect on ZCTG, we interchange the LLM

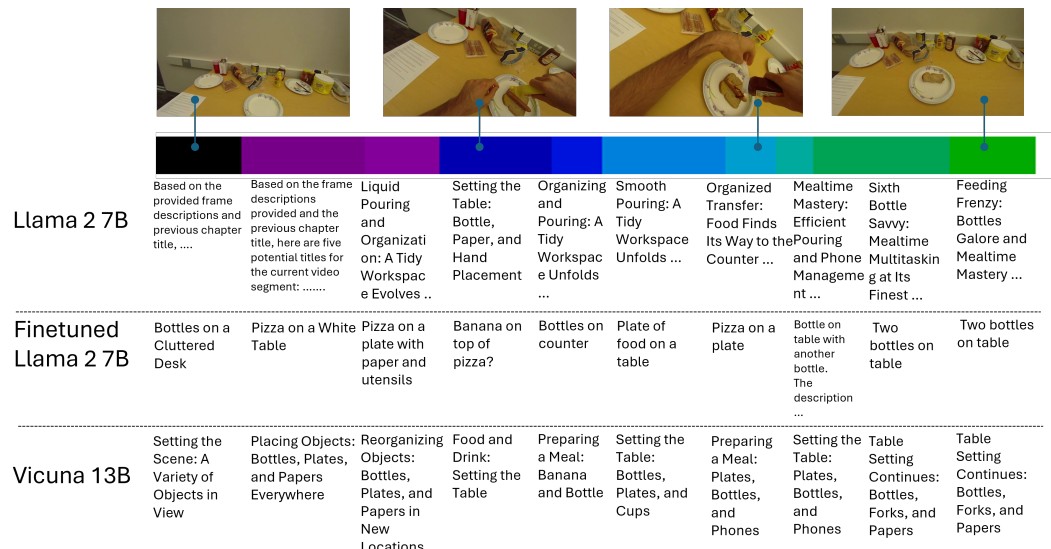

| | | | | | | | | | |
|---|---|---|---|---|---|---|---|---|---|
| **Llama 2 7B** | Based on the provided frame descriptions and previous chapter title, .... | Based on the frame descriptions provided and the previous chapter title, here are five potential titles for the current video segment: ....... | Liquid Pouring and Organization: A Tidy Workspace Evolves .. | Setting the Table: Bottle, Paper, and Hand Placement | Organizing and Pouring: A Tidy Workspace Unfolds ... | Smooth Pouring: A Tidy Workspace Unfolds ... | Organized Transfer: Food Finds Its Way to the Counter ... | Mealtime Mastery: Efficient Pouring and Phone Management ... | Sixth Bottle Savvy: Mealtime Multitasking at Its Finest ... | Feeding Frenzy: Bottles Galore and Mealtime Mastery ... |
| **Finetuned Llama 2 7B** | Bottles on a Cluttered Desk | Pizza on a White Table | Pizza on a plate with paper and utensils | Banana on top of pizza? | Bottles on counter | Plate of food on a table | Pizza on a plate | Bottle on table with another bottle. The description ... | Two bottles on table | Two bottles on table |
| **Vicuna 13B** | Setting the Scene: A Variety of Objects in View | Placing Objects: Bottles, Plates, and Papers Everywhere | Reorganizing Objects: Bottles, Plates, and Papers in New Locations | Food and Drink: Setting the Table | Preparing a Meal: Banana and Bottle | Setting the Table: Bottles, Plates, and Cups | Preparing a Meal: Plates, Bottles, and Phones | Setting the Table: Plates, Bottles, and Phones | Table Setting Continues: Bottles, Forks, and Papers | Table Setting Continues: Bottles, Forks, and Papers |

Figure 5: Generated chapters and their titles from ZCTG using different LLM models for a video of making a hotdog from GTEA.

block with various LLMs while keeping all other elements same. This allows us to evaluate how different factors, such as the pretrained knowledge and size of the LLM, influence the generated titles.

Table 4 contains the evaluation results for three LLM models: Llama 2 (7B) Touvron et al. (2023), a fine-tuned version of Llama 2 (7B) on the R-VQA Lu et al. (2018) dataset, and Vicuna v1.5 (13B) Zheng et al. (2023). For more details on the fine-tuning process for Llama 2, please refer to A.5. Figure 5 illustrates the generated titles for a video of hotdog preparation from the GTEA dataset, using different LLMs. We observe that the Vicuna v1.5 model consistently produced the best results. In contrast, the titles generated by Llama 2 tend to be excessively lengthy and often fail to accurately reflect the

Table 3: Evaluation results using Judge LLM (ChatGPT- 3.5) on VidChapters-7M dataset.

| Criterion | Ours |
|---|---|
| Contextual understanding | 1.63 |
| Temporal understanding | 0.77 |
| Correctness of information | 0.45 |

video content. Although the titles from the fine-tuned Llama 2 are more concise, they sometimes include inaccuracies, such as mentioning a 'banana on top of pizza'. This experiment highlights that larger models (13B compared to 7B here), which incorporate additional knowledge, tend to yield superior results.

Table 4: Evaluation results when different LLM models are used in ZCTG on GTEA dataset.

| LLM | CS |
|---|---|
| Llama 2 (7B) | 14.01 |
| Fine-tuned Llama 2 (7B) | 21.08 |
| Vicuna 1.5 (13B) | 25.40 |

Video chapter generator evaluation: Segmenting videos into chapters often requires prior knowledge of the true number of segments, a requirement even for many existing unsupervised methods Sarfraz et al. (2019; 2021). However, this assumption may not always be practical. That is why we refrain from assuming any such prior knowledge. Nonetheless, for our chapter generation module, we use TW-FINCH, which necessitates defining the number of clusters beforehand. On average, videos on platforms like YouTube typically comprise 10-15 segments, each with distinct semantics. It is important to note that the number of clusters does not necessarily equate to the number of chapters. A cluster can encompass one or more video chapters. Thus, we set $K = 10$.

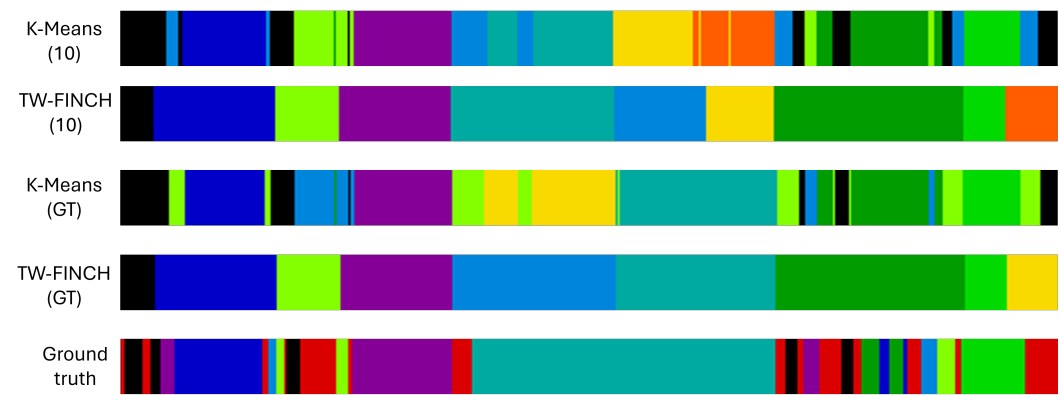

Figure 6: Video Chapter Generation by segmenting temporally using K-Means and TW-FINCH for a video of making tea from GTEA dataset.

Intuitively, well-constructed chapters should yield superior titles. To examine this, we compare the video chapters generated by K-Means and TW-FINCH. The results are presented in Table 5 and an example of video chapters or segments for a video of making tea from GTEA dataset is shown in Figure 6. We use the Hungarian matching algorithm to match the generated segments and ground-truth to calculate the metrics. We observe that TW-FINCH achieves higher scores compared to K-Means. This can be attributed to the temporal weighting in TW-FINCH, which mitigates over-segmentation and outperforms K-Means.

Table 5: Evaluation of algorithms in Video Chapter Generator on GTEA dataset. *The number of clusters is set to ground-truth number of clusters for every video.

| Method | K | MoF | IoU |
|---|---|---|---|
| K-Means | 10 | 22.46 | 0.121 |
| TW-FINCH | 10 | 26.47 | 0.155 |
| K-Means | GT* | 27.72 | 0.157 |
| TW-FINCH | GT* | 29.98 | 0.177 |

Varying $Q$ from Scene Graph Generator: The Scene Graph Generator plays a vital role in ZCTG by transforming visual content into textual input suitable for LLM interpretation. We used a pre-trained module Li et al. (2021a) as our Scene Graph Generator. To examine the effect of the amount of information from the scene graph given as input to LLM on the final results, we experiment with different values of $Q$, which represents the number of subject-object triplets included in the LLM input. The results, including evaluation scores when $Q$ is varied, and an example of generated titles is in A.1. We observe that the best performance is achieved when $Q = 10$, which is the value used in all our experiments. This indicates that a very low or high value of $Q$ can reduce the quality of generated titles.

## 5 LIMITATIONS AND FUTURE WORK

We introduced a novel zero-shot framework, ZCTG, designed to simplify video content navigation by generating video chapters and their corresponding titles. While ZCTG demonstrates strong capabilities in generating chapter titles that align closely with visual content in a zero-shot setting, it has certain limitations. One limitation is relying only on visual features to create video chapters, which can often result in oversegmentation. A promising future work to address this issue is refining chapter boundaries using semantic information from scene graphs.

Although ZCTG integrates temporal information at multiple steps in the framework, it does not always capture and reason about specific actions in videos, partly due to limited context from scene graphs. A future direction would be to leverage LLMs to enhance both spatial and temporal context, thereby improving the quality of the generated titles. We envision ZCTG to help advance research in video comprehension, especially in the genre of video chapter generation.

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

# A APPENDIX

We present the following in this Appendix section:

## A.1 VARYING $Q$ EXPERIMENT RESULTS

We present the results for varying $Q$, number of triplets considered from Scene Graph Generator. Table 6 shows the CLIPScore when value of $Q$ is varied. We find that $Q = 10$ yields the best score.

To further support this finding, we present an example of generated titles for a video on hotdog preparation from the GTEA dataset, illustrating how different values of $Q$ affect output quality. These results indicate that using significantly fewer or more triplets (as in the cases of $Q = 5$ or $Q = 20$) leads to lower quality titles and a decline in overall performance.

Table 6: Results for varying $Q$ values from Scene Graph Generator module on GTEA dataset.

| Q | CS |
|---|-----|
| 5 | 25.32 |
| 10 | 25.40 |
| 15 | 25.28 |
| 20 | 25.18 |

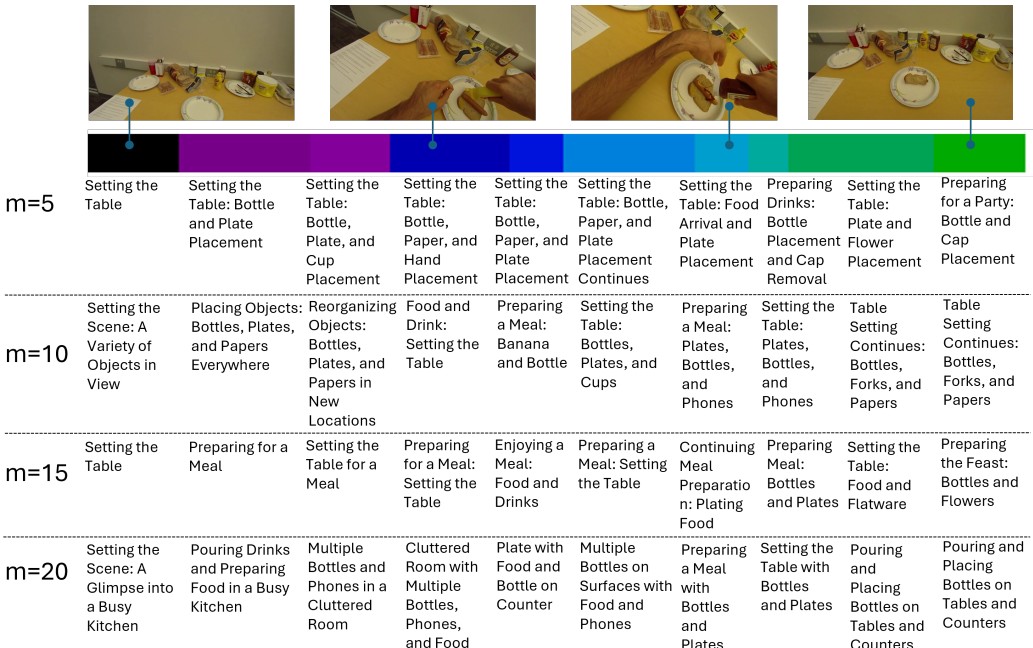

Figure 7: Generated chapter titles using ZCTG for a video of making a hotdog from the GTEA dataset when the number of triplets from the Scene Graph Generation module is varied.

## A.2 LLM-BASED EVALUATION

We adapt the evaluation prompts used by Maaz et al. (2023) with Judge LLM, ChatGPT 3.5. As previously mentioned, due to the zero-shot nature of our framework, the number of ground-truth titles may differ from the generated titles because of the varying number of video chapters. To address this discrepancy, we employ the following evaluation strategy: for each ground-truth segment and its corresponding title, we include all predicted titles for that segment when calculating the evaluation metrics.

For instance, if a ground-truth segment has title 'A' in the range $\{s_1, s_2\}$, and our framework predicts three segments in this range with titles 'B', 'C', and 'D', we compare as follows: Ground-truth = 'A' and Predictions = 'B', 'C', 'D'. For metrics requiring one-to-one comparisons, 'A' will be compared individually with 'B', 'C', and 'D', and the average metric value will be calculated.

Following are the prompts used for each of the three aspects of this evaluation:

### Contextual understanding

**System Prompt**:

```
You are an intelligent chatbot designed for evaluating the contextual
understanding of generative outputs for video-based chapter titles. Your
task is to compare the predicted chapter title with the correct title and
 determine if the generated response aligns with the overall context of
the video content. Here's how you can accomplish the task:
------
##INSTRUCTIONS:
- Evaluate whether the predicted chapter aligns with the overall context
of the video segment content. The content can be inferred from the video
title marked as Correct Answer. It should not provide information that is
 out of context or misaligned.
- The predicted answer must capture the main themes and sentiments of the
 video. If the predicted answer is able to capture the objects in the
segment its score should be less than the scenario where it detects
objects as well as the interaction between them (actions).
- Consider synonyms or paraphrases as valid matches.
- Provide your evaluation of the contextual understanding of the
prediction compared to the answer.
```

**User Prompt**:

```
Please evaluate the following video chapter titles:
Correct Answer: {Ground-truth}
Predicted Answer: {Predictions}
Provide your evaluation only as a contextual understanding score where
the contextual understanding score is an integer value between 0 and 5,
with 5 indicating the highest level of contextual understanding. Please
generate the response in the form of a Python dictionary string with keys
 'score', where its value is contextual understanding score in INTEGER,
not STRING. DO NOT PROVIDE ANY OTHER OUTPUT TEXT OR EXPLANATION. Only
provide the Python dictionary string. For example, your response should
look like this: {''score': 4.8}.
```

### Correctness of information

**System Prompt**:

```
You are an intelligent chatbot designed for evaluating the factual
accuracy of generative outputs for video-based chapters.
Your task is to compare the predicted answer with the correct answer and
determine if they are factually consistent. Here's how you can accomplish
 the task:
------
##INSTRUCTIONS:
- Focus on the factual consistency between the predicted answer and the
correct answer. The predicted answer should not contain any
misinterpretations or misinformation.
```

```
- The predicted answer must be factually accurate and align with the
video content.
- Consider synonyms or paraphrases as valid matches.
- Evaluate the factual accuracy of the prediction compared to the answer.
```

**User Prompt**:

```
Please evaluate the following video chapters:
Correct Answer: {Ground-truth}
Predicted Answer: {Predictions}
Provide your evaluation only as a factual accuracy score where the
factual accuracy score is an integer value between 0 and 5, with 5
indicating the highest level of factual consistency.
Please generate the response in the form of a Python dictionary string
with keys 'score', where its value is the factual accuracy score in
INTEGER, not STRING.
DO NOT PROVIDE ANY OTHER OUTPUT TEXT OR EXPLANATION. Only provide the
Python dictionary string.
For example, your response should look like this: {'score': 4.8}.
```

**Temporal understanding**

**System Prompt**:

```
You are an intelligent chatbot designed for evaluating the temporal
understanding of generative outputs for video-based chapters.
Your task is to compare the predicted answer with the correct answer and
determine if they correctly reflect the temporal sequence of events in
the video chapter's content. Here's how you can accomplish the task:
------
##INSTRUCTIONS:
- Focus on the temporal consistency between the predicted answer and the
correct answer. The predicted answer should correctly reflect the
sequence of events or details as they are presented in the video content.
- Consider synonyms or paraphrases as valid matches, but only if the
temporal order is maintained.
- Evaluate the temporal accuracy of the prediction compared to the answer
.
```

**User Prompt**:

```
Please evaluate the following video chapters:
Correct Answer: {Ground-truth}
Predicted Answer: {Predictions}
Provide your evaluation only as a temporal accuracy score where the
temporal accuracy score is an integer value between 0 and 5, with 5
indicating the highest level of temporal consistency.
Please generate the response in the form of a Python dictionary string
with keys 'score', where its value is the temporal accuracy score in
INTEGER, not STRING.
DO NOT PROVIDE ANY OTHER OUTPUT TEXT OR EXPLANATION. Only provide the
Python dictionary string.
For example, your response should look like this: {'score': 4.8}.
```

## A.3 SUMMARIZATION PROMPT FOR VIDCHAPTERS-7M

To generate a chapter title, we first aggregate the descriptions of all frames within the chapter. However, for long videos, such as those in the VidChapters-7M dataset, the volume of frame descriptions often exceeds memory and context length limits. To manage this, we summarize the descriptions every 20 frames. We chose this interval to balance between minimizing information loss and staying within memory constraints. These summarized descriptions are then aggregated to generate the chapter title. For example, if a chapter contains 100 frames, instead of aggregating 100 individual descriptions, we use 5 summarized frame descriptions. A straightforward summarization prompt (shown below) is used for this intermediate step.

**System Prompt**:

```
Provide a concise summary (in less than 50 words) in one sentence for the
 following frame descriptions:
```

**User Prompt**:

```
{list of frame descriptions}
```

### A.4   VIDEOLLAMA EVALUATION ON VIDCHAPTERS-7M

Due to VideoLlama's inability to process lengthy videos and memory limitations, we evaluate both VideoLlama and ZCTG on a subset of 50 randomly selected videos from VidChapters-7M. In this subset, the number of frames ranges from 100 to 700 when sampled at 1 FPS. With an FPS typically ranging between 24 and 60, this subset of videos accurately reflects the average length of videos across the entire dataset. ZCTG achieves a CLIPScore of 21.3, outperforming VideoLlama, which scores 17.3.

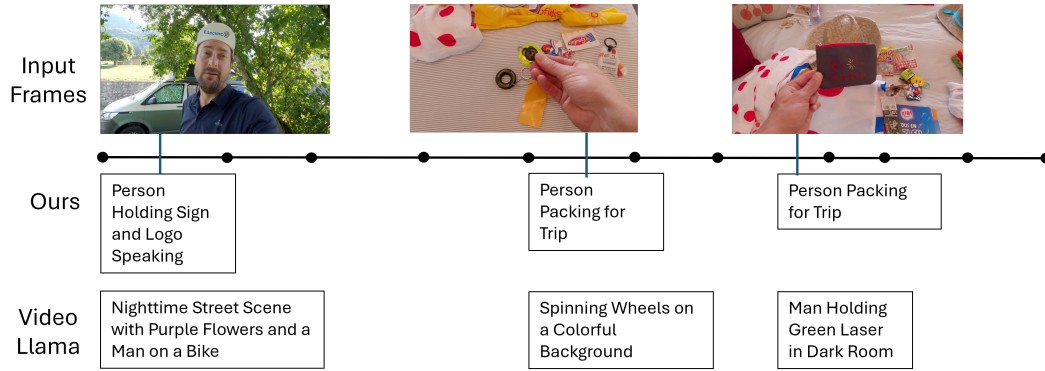

Figure 8: Generated chapter titles by ZCTG and VideoLlama for a video about Tour de France from VidChapters-7M dataset.

We provide an example of generated titles in Figure 8. It is clear that the titles generated by VideoLlama do not capture the spatio-temporal cues very well. For example, the second title displayed describes the spinning wheel in the frame but fails to capture the broader context, whereas the title generated by ZCTG, 'Person packing for the trip,', captures the ongoing activities in the chapter accurately.

### A.5   FINE-TUNING LLAMA 2

In order to examine the effect of fine-tuning LLM on the generated chapter titles, we fine-tune Llama 2 (7B) model. Since ZCTG does not have access to ground-truth titles and there are no frame-level descriptions available, fine-tuning on either the GTEA or VidChapters-7M datasets is not feasible. Hence, we opted for the Relation-VQA (R-VQA) dataset Lu et al. (2018) for this task. The R-VQA dataset is derived from the Visual Genome (VG) dataset and includes a question, its correct answer, and a supporting fact in the form of an object-relation triplet. We selected this dataset because it includes supporting facts in the form of object relations, which closely aligns with the scene graph information utilized in our task. Below is a sample input and the expected response from the dataset:

```
Below is an instruction that contains a question, paired with input that
provides context in the form of <subject, relation, object>. Write a
response that provides appropriate answer to the question.### Question:
What white lines are in the background?
### Input:
lines, are, white
### Response:
Crosswalk lines.
```

For fine-tuning the Llama 2 model, we use the Low-Rank Adaptation technique (LoRA) technique and HuggingFace Wolf et al. (2019) library. The base Llama 2 model is trained for 200 iterations with an initial learning rate of 0.0002. The objective is to answer the question using the provided supporting object-relation triplet.

We use the train, validation, and test splits provided by the authors for fine-tuning. Specifically, the training set comprises 119,333 samples, the validation set includes 39,777 samples, and the test set contains 39,779 samples. We observed that the fine-tuned Llama 2 model generated shorter and descriptive titles compared to the base Llama 2, an observation reflected in the final results (refer to Figure 5).

## A.6 LLM PROMPTS FOR ZCTG

Here, we show the prompts used for our experiments. We use the same prompts for both datasets. After multiple prompt optimization iterations, we use the following system and user prompt for generating the frame descriptions $d_i$.

**System Prompt**:

```
You are a prompt engineer trying to optimize the text description of a
video action for action segmentation. You are given a list of triplets
where each triplet is in the format of {id1_object => action =>
id2_object}. Here "action" represents the interaction between the objects
 "id1_object" and "id2_object". The list of triplets indicates the
actions taking place in a given video frame (or set of video frames).
Additionally, you will be provided with a previous frame description to
guide the description generation. Your goal is to optimize the
description for the given list of actions and the previous frame
description that uniquely identifies what is happening in the video and
where it is taking place.
Some tips to optimize the description:
1. Use the causal nature of physical events to predict the main action
for the given list. For example, bottle in hand can refer to several
actions, such as pouring out of the bottle, closing the cap on the bottle
, etc.
2. Each object is preceded by a number, identifying it as a different
category. Objects with the same number are the same objects, and vice
versa. For example, 1_bottle and 2_bottle refer to two different bottles
in the same scene. The description should not confuse the reader into
thinking they are the same bottle.
3. Please use the previous frame description as a reference to predict
what is happening in the scene and guide the description generation
process.
```

**User Prompt**:

```
Shared below is a list of triplets that represent the scene graph of a
video frame and the previous frame description. Please provide a short
description (strictly within 15 words) to describe the events or actions
happening in the frame.
```

To generate the video chapter titles, we use the following system and user prompts for our experiments.

**System Prompt**:

```
You are a video annotator who is tasked to generate a single title given
a video segment information. The information is given as {<frame_desc>; <
prev_chap>} where frame_desc is a list of descriptions of events in the
set of frames in the current segment, and prev_chap is the chapter title
generated for the previous video segment. The list of descriptions
indicates the actions taking place in a given video frame (or set of
video frames). The prev_chap title is an indication of the flow of the
sequence of actions in the video. Please ensure that the action taking
place in the segment (for example, eating, drinking, running, etc.) is
mentioned in the title.
```

**User Prompt**:

```
Below is a list of frame descriptions (<frame_desc>) and the title for
the previous video segment (<prev_chap>). Please generate an appropriate
title (STRICTLY less than 20 words) for the corresponding video clip
using the scene description given in frame_desc and prev_chap as a
reference. DO NOT copy the prev_chap (literally and semantically). For
context, <prev_chap> denotes the actions that took place just before this
 video segment. So please try to consider the sequence of actions (causal
 nature of physical events), the current frame description and previous
segment chapter title, and predict what is taking place in the current
video segment. Generate a title based on that information.
```

A.7  PROMPTS FOR VIDEOLLAMA

For Video-Llama experiments, we use the below system and user prompts. While these prompts retain the core essence of those utilized in ZCTG experiments, they are subtly adjusted to maximize the performance of the Video-Llama model. For instance, the inclusion of the phrase 'DO NOT ADD any additional text like Sure! or Certainly! in your response.' became necessary due to frequent additions of such text, which is undesired in the chapter titles.

**System Prompt**:

```
You are a video annotator who is tasked to generate a single title for a
video segment or clip. Please generate an appropriate title (STRICTLY
less than 20 words) for the corresponding video clip using your ability
for scene understanding. DO NOT ADD any additional text like Sure! or
Certainly! in your response. The output only needs to be a title (less
than 20 words).
```

**User Prompt**:

```
Please provide a chapter title (STRICTLY less than 20 words) for the
provided video segment. DO NOT describe the scene in detail.
```

A.8  MORE EXAMPLES FOR ZCTG

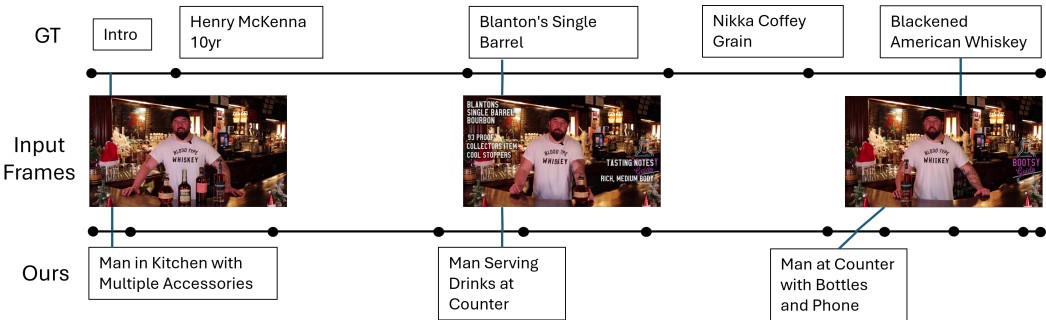

Figure 9: Generated titles by ZCTG and ground-truth for a video about whiskey gift guide from VidChapters-7M dataset.

