# OpenReview forum: "ZCTG: A Zero-Shot Framework for Automatic Video Chaptering and Title Generation"
_ICLR.cc/2025/Conference — ICLR 2025 Conference Withdrawn Submission_

### Official Review · Reviewer_7STg · 2024-10-21

**Soundness:** 3
**Presentation:** 3
**Contribution:** 2
**Rating:** 5
**Confidence:** 4

**Summary:**

This paper focuses on the task of automatic video chapter generation and title generation and introduces a method named Zero-Shot Video Chapter Title Generator (ZCTG). The proposed framework consists of two primary tasks *video chapter generation* and *chapter title generation*, which are solved with the pretrained modules including *scene graph generator*, *spatio-temporal feature extractor*, *video chapter generator*, and *large language model*. The experiments are conducted on VidChapters-7M and GTEA datasets.

**Strengths:**

1. The paper is easy to follow.
2. To the best of my knowledge, the proposed method is the first one for automatic video chapter and title generation in a zero-shot scenario.

**Weaknesses:**

1. My biggest concern is the innovation of the paper. Since the task of video chapter title generation has been defined and studied by VidChapters-7M, it is not the first work to solve this problem. Besides, all modules in this framework are taken from other works, such as I3D, TW-FINCH, and Vicuna v1.5, so this work is more of an engineering one than an acedemic one.
2. It would be better if the authors can explore more possibilities by trying other spatio-temporal feature extractor and scene graph generator.
3. It seems that the authors did not report the MoF and IoU metrics on VidChapters-7M.

**Questions:**

How do the authors view this task as different from other tasks such as video captioning and grounding? If there is a strong video captioning model and video grounding model, is it possible to segment the video by segmenting the captions?

---

### Official Review · Reviewer_5kiY · 2024-10-28

**Soundness:** 2
**Presentation:** 3
**Contribution:** 2
**Rating:** 5
**Confidence:** 4

**Summary:**

This paper proposes a zero-shot framework for automatically generating chapter segments and titles for videos relying solely on visual information from video frames (no speech transcript or audio). ZCTG first segments the video into chapters using an unsupervised temporal action segmentation technique. Then, for each chapter, it extracts scene graphs from the frames, converts these into textual descriptions using an LLM, and finally uses another LLM to generate a chapter title based on these descriptions and the title of the preceding chapter to ensure coherence.

Experiments on the VidChapters-7M and GTEA datasets show that ZCTG can generate chapters in a zero-shot manner better than a few baselines. The paper also discusses the impact of different LLMs and the number of scene graph triplets used on the quality of generated titles.

**Strengths:**

- The method can chapter videos without requiring training on annotated chapter data and relying on visual cues solely.

**Weaknesses:**

- Are scene graph based captions better for video chaptering than frame captions extracted directly by a state-of-the-art VLM?
- The CLIPScore metric is used throughout the paper, but it is not clear that it is well suited to evaluate chapter titles. For instance, the VidChapters-7M paper mentions over 25% of videos have their chapters speech-only based, hence CLIPScore is unlikely to accurately measure the quality of chapter titles for these videos. Have these videos been excluded, or has any study been conducted to evaluate the relevance of CLIPScore for this purpose?
- Table 3 with LLM evaluation of chapters lacks information. How do these numbers compare to baselines (e.g. Vid2Seq zero-shot and finetuned)? Also, while LLM for QA evaluation has been widely studied, it is not trivial that LLM do a good job at evaluating chapter segmentation / timestamp outputs. Has any study been conducted to measure this?
- The paper does not report the chapter segmentation metrics reported in the VidChapters paper. It would be beneficial to have them for comparison.

**Questions:**

See above.

---

### Official Review · Reviewer_D3jo · 2024-11-03

**Soundness:** 2
**Presentation:** 2
**Contribution:** 2
**Rating:** 3
**Confidence:** 3

**Summary:**

The ZCTG (Zero-Shot Chapter Title Generator) framework is proposed for automatic video chaptering and title generation using only video data without the need for annotations or additional input modalities. It leverages scene graphs and Large Language Models (LLMs) to achieve efficient, zero-shot chapter generation and titling across various video types. The framework shows competitive results in generating concise, contextually accurate chapter titles using visual information alone, as demonstrated on datasets like VidChapters-7M and GTEA.

**Strengths:**

1. This work provides a zero-shot solution for this task.
2. The experimental results show the superiority of the proposed method.

**Weaknesses:**

1. Presentation is not clear.
	1. What is the dot line in Figure 1? Figure 1 needs a caption.
2. Lack of ablations.
	1. How about using other LLMs, Video chapter generators, and spatial-temporal feature extractor?
	2. How about directly generate captions for each frame and summarize them? Will this work better?
	3. How do different prompts affect the performance?
3. Lack of novelty. This paper seems to directly combine different existing models. Did not see special design for this task or addressing the hallucination of LLMs and scene graph generator.

**Questions:**

See weakness

---

### Official Review · Reviewer_KXZD · 2024-11-04

**Soundness:** 3
**Presentation:** 3
**Contribution:** 2
**Rating:** 3
**Confidence:** 4

**Summary:**

This work presents ZCTG, a zero-shot framework for generating video chapters and their titles for untrimmed videos. ZCTG combines scene graphs and Large Language Models (LLMs) to create chapters without relying on additional data modalities or detailed supervision. The framework operates solely on video data and does not require any training. An evaluation on the VidChapters-7M and GTEA datasets demonstrates the effectiveness of this approach across different video durations and domains

**Strengths:**

The paper introduces a zero-shot framework for video chaptering and title generation, which is a significant departure from traditional methods that rely heavily on annotated data.

By leveraging scene graphs and large language models, the approach creatively combines existing technologies to address the problem in a new way.

**Weaknesses:**

The framework creatively combines existing technologies such as scene graphs and large language models (LLMs). While this integration is innovative, it is an incremental improvement.

There are several baselines in the paper VidChapter7M, which have better results (Table 3 first part as without fine-tuned) than the reported results in the paper (Table 1) on (B1 to B4 metric).

The integration of scene graphs and large language models (LLMs) can lead to high computational demands. This might limit the framework’s accessibility and scalability.

VidChapter7M: https://arxiv.org/pdf/2309.13952

**Questions:**

Any failure cases where the framework is not able to perform well.

There is no use of other modalities (speech), as experimented in VidChapter7M, which improves the performance. Why it is not included in this study?

---

### Note · Authors · 2024-11-19

I have read and agree with the venue's withdrawal policy on behalf of myself and my co-authors.